# Study of the Antibacterial Capacity of a Biomaterial of Zeolites Saturated with Copper Ions (Cu^2+^) and Supported with Copper Oxide (CuO) Nanoparticles

**DOI:** 10.3390/nano13142140

**Published:** 2023-07-24

**Authors:** Lina M. Romero, Nicolas Araya, Daniel A. Palacio, Gabriela A. Sánchez-Sanhueza, Eduardo G. Pérez, Francisco J. Solís, Manuel F. Meléndrez, Carlos Medina

**Affiliations:** 1Interdisciplinary Group of Applied Nanotechnology (GINA), Hybrid Materials Laboratory (HML), Department of Materials Engineering (DIMAT), Faculty of Engineering, Universidad de Concepción, 270 Edmundo Larenas, Box 160-C, Concepcion 4070409, Chile; lromerocardenas89@correo.unicordoba.edu.co (L.M.R.); nicolasiaraya@udec.cl (N.A.); mmelendrez@udec.cl (M.F.M.); 2Departamento de Polímeros, Facultad de Ciencias Químicas, Universidad de Concepción, 129 Edmundo Larenas, Concepcion 4070409, Chile; dapalacio@udec.cl; 3Department of Restorative Dentistry, Faculty of Dentistry, Universidad de Concepción, Concepcion 4070409, Chile; gasanchez@udec.cl; 4Facultad de Ciencias Físico-Matemáticas, Universidad Autónoma de Nuevo León, San Nicolas de los Garza, Nuevo Leon 66451, Mexico; eduardo.pereztj@uanl.edu.mx (E.G.P.); francisco.solispm@uanl.edu.mx (F.J.S.); 5Unidad de Desarrollo Tecnológico, 2634 Av. Cordillera, Parque Industrial Coronel, Box 4051, Concepcion 4191996, Chile; 6Department of Mechanical Engineering (DIM), Faculty of Engineering, Universidad de Concepción, Edmundo Larenas 270, Concepcion 4070409, Chile

**Keywords:** zeolites, copper ions, antibacterial, nanoparticles

## Abstract

In this work, copper (II) ions were saturated and copper oxide nanoparticles (CuO NPs) were supported in natural zeolite from Chile; this was achieved by making the adsorbent material come into contact with a copper ion precursor solution and using mechanical agitation, respectively. The kinetic and physicochemical process of the adsorption of copper ions in the zeolite was studied, as well as the effect of the addition of CuO NPs on the antibacterial properties. The results showed that the saturation of copper (II) ions in the zeolite is an efficient process, obtaining a 27 g L^−1^ concentration of copper ions in a time of 30 min. The TEM images showed that a good dispersion of the CuO NPs was obtained via mechanical stirring. The material effectively inhibited the growth of Gram-negative and Gram-positive bacteria that have shown resistance to methicillin and carbapenem. Furthermore, the zeolite saturated with copper at the same concentration had a better bactericidal effect than the zeolite supported with CuO NPs. The results suggested that the ease of processing and low cost of copper (II) ion-saturated zeolitic material could potentially be used for dental biomedical applications, either directly or as a bactericidal additive for 3D printing filaments.

## 1. Introduction

Recently, minerals that present high porosity, such as zeolite, have become a useful and economical alternative for the development of biomaterials with unique physical and chemical properties for numerous applications. The incorporation of metallic nanostructures with bactericidal properties can provide solutions in human medicine [1]. The diversity of applications is due, specifically, to the porosity of this material, which has a structure with negatively charged cavities that make it a striking material due to the exchange of monovalent and divalent ions, the presence of hydroxyl groups and water molecules [2], as well as the lodging of other molecules in their cavities, such as ammonia and nitrates [3].

Clinoptilolite-type zeolites are one of the most abundant and important natural zeolites in applications in humans, veterinary medicine, and environmental applications [4]. These are aluminosilicate minerals with tetrahedral anionic frameworks with well-defined channels and cavities which can be occupied by water molecules or compensating cations. These neutralize the anionic charges in the cavities and in the formation of tetrahedral rings [5]. Most types of zeolites are of volcanic origin and have the general formula M_2_/n:Al_2_O_3_:xSiO_2_:yH_2_O, where M is the cation found in the cavities or pores of the material, with the main structure being based on tetrahedrons. AlO_4_ and SiO_4_ share between one and three oxygen atoms, so their structure can vary as they extend in three dimensions [6].

In the search for applications at the microbiological level, both organic and inorganic materials have been considered, such as clay materials [7], nanocellulose fibers, and carbon nanotubes, among other types of materials [8,9,10,11]. In this context, Cu or CuO nanostructures have been highlighted as promising nanomaterials with high antibacterial capacity and great potential to be used against a wide variety of Gram-positive and Gram-negative bacteria [12,13,14], where the mechanism of action for the bacteria is due to the reaction that these metals cause when combined with the SH groups of enzymes that inactivate cellular proteins [15]. Lei et al. [16] reported that KTO nanowires doped with Cu and Ag had enhanced biological activity, such as antibacterial properties. The nanomaterial developed solely with copper (1.0 Cu-KTO) obtained the same inhibition of the bacterial population as the nanomaterials doped with both metals, indicating the high bactericidal activity of copper and inspiring a new idea for the design and development of new antibacterial nanomaterials.

Rodríguez-Méndez et al. [17] studied the formation of silver nanoparticles supported on a natural zeolite and evaluated their antimicrobial effect. The results showed that the nanoparticles were well dispersed and stable and had a high bactericidal effect on large populations of bacteria. Yao et al. [18] studied the antimicrobial capacity of zeolites exchanged with Cu^2+^ and Zn^2+^ ions on *Escherichia coli* and *Staphylococcus aureus*, demonstrating that the cations of the respective metals were housed on the surfaces and cavities of the zeolites through ionic exchange processes, as it was also demonstrated that saturated zeolites with Cu^2+^ ions show excellent antibacterial performance with *S. aureus*, reaching 100% mortality rates with 1000 mg L^−1^ concentrations of Cu^2+^ after 1 h. Cruces et al. [12] studied the effect of bimetallic nanoparticles of copper/silver supported in geoaluminum materials such as antibacterial agents, finding a homogeneous distribution on the surface of the geomaterials and demonstrating that the use of geomaterials with nanoparticle supports increases their contact surfaces, which is favored in a greater release of ions and directly improves the antibacterial activity, showing the great advantage of using these aluminosilicate materials as antibacterial agents. Chen et al. [19] studied the ion release properties and their antibacterial efficiency for saturated zinc, copper, and iron, observing a greater release of copper and zinc ions in the presence of a saline solution, with a 73% release rate for zinc and 36% for copper and a minimum bactericidal concentration after 2 h of 32 μg mL^−1^ and 64 μg mL^−1^ for copper and zinc, respectively.

The objective of this research is focused on obtaining an antimicrobial biomaterial from a natural zeolite saturated with Cu^2+^ ions and supported with copper oxide nanoparticles, taking advantage of the high surface area of the zeolite that will allow for the dispersion of CuO nanoparticles and a suitable antibacterial performance. Furthermore, natural zeolites have a great advantage due to their high availability, low cost, affinity for cation exchange, and non-toxic nature, making them an efficient antimicrobial biomaterial. Concentrations of zeolites saturated with Cu^2+^ and CuO were evaluated and used to generate different discs; they were then used as antibacterial biomaterials against strains multi-resistant to antibiotics selected from the dental biomedical field.

## 2. Materials and Methods

### 2.1. Reagents

Natural zeolite (ZLn) was obtained from Zeomaule (Maule, Chile). Hydrochloric acid (37 wt%) and pentahydrate copper sulfate (99.8%) were obtained from Sigma-Aldrich (Santiago, Chile). Copper oxide (CuO) nanoparticles were supplied by US Research Nanomaterials Inc. (Houston, TX, USA). The reagents used in the microbiological tests were selected based on the recommendations of the Clinical and Laboratory Standards Institute (CLSI/M100-S30) [20], and modifications to the testing protocol were made as described by [21,22,23].

### 2.2. Zeolite Washing 

Washing was performed to remove any residual organic matter from the extraction process and reduce the presence of iron, following the method described by [24]. In a round-bottom flask, a ratio of 1.0 g of ZLn per 100 mL of hydrochloric acid (2.4 mol L^−1^) was added. The solution was constantly agitated at 200 rpm for 24 h at room temperature. After 24 h, the excess hydrochloric acid was eliminated by thoroughly rising with water and centrifuging the sample at a (>9000 rpm) for 10 min. The resulting sample was dried for 24 h at 65 °C.

### 2.3. Saturated Zeolite with Copper Ions 

The saturation tests of the washed natural zeolite (ZLnL) aimed to determine the optimal conditions for exchanging copper ions. Three variables were assessed: zeolite content, copper concentration, and contact time. To evaluate these parameters, a specific amount of zeolite was exposed to 15 mL of copper solution derived from pentahydrate copper sulfate salt (CuSO_4_⋅5H_2_O, 99.8%). The system was continuously stirred at 200 rpm for 24 h. (i) The zeolite dose was varied (0.1, 0.25, 0.5, 0.75, 1.0, and 2.0 g) while maintaining a constant concentration of copper ions (50 g L^−1^). (ii) To assess the effect of copper concentrations, the zeolite dose remained constant, and the concentration of copper ions was varied (5.0, 10.0, 15.0, 20.0, 30.0, 40.0, and 50.0 g L^−1^). (iii) for the variation in time, different contact times (0.5 h to 24.0 h) were tested at a predetermined dose of zeolite and copper concentration. After each trial, the obtained material was centrifuged (>10,000 rpm) for 15 min. The solution was analyzed using atomic adsorption spectroscopy to determine the remaining copper contents. The saturated zeolite was stored at 60 °C for 24 h and passed through a mesh with a size of (125 μm). It was finally stored in hermetic containers for subsequent analysis and characterization. Henceforth, to clarify the identifications of the material obtained, it will be referred to as saturated zeolite with copper ions (ZLnLCu).

### 2.4. Zeolite Supported with Copper Oxide Nanoparticles 

The zeolite surface was coated with nanoparticles through mechanical agitation. This process involved taking the previously prepared ZLnLCu (27.0 g L^−1^) and 63.0 g L^−1^ of pre-synthesized copper oxides nanoparticles (CuO NPs) with a size range of 20–100 nm. The aim was to achieve a maximum concentration of copper ions and nanoparticles at 100.0 g L^−1^. To achieve this, the ZLnLCu solution was mixed with a specific amount of CuO NPs and homogenized using an agate mortar until a homogeneous mixture was obtained. The resulting material will be referred to as ZLnLCuONPs.

### 2.5. Zeolite Characterization 

The ZLn was structurally characterized by X-ray diffraction using a D4 ENDEAVOR XRD diffractometer (Bruker, MA, USA). The characterizations of the ZLn, ZLnL, ZLnLCu, and ZLnLCuONPs were first performed by infrared spectroscopy (FTIR) with the attenuated total reflection function (ATR) (FTIR-ATR) using a FTIR Spectrum Two spectrometer (×1720) (Perkin Elmer, Waltham, MA, USA). Scanning electronic microscopy (SEM) images were taken using a model JSM– 6300 scanning electron microscope from JEOL, MA, USA ( with 20 kV acceleration voltage, and the samples were also analyzed via X-ray spectroscopy of dispersive energy (EDS). Dee transmission electronic microscopy images were performed using a JEM 1200 ex II Ex-transmission microscope (JEOL, Ltd., Tokyo, Japan) at a voltage of 120 kV. Selected area electron diffraction (SAED) was also carried out in TEM in order to clarify the crystalline structure and to further highlight the presence of CuO particles in the composite product.

### 2.6. Microbiological Assays 

In a first step, the antibacterial activity of all modified zeolites against the clinical methicillin-resistant *Staphylococcus aureus* [25] strain and American-Type Control Culture ATCC *Enterococcus faecalis* 29212 strain was determined by minimum inhibitory concentration (MIC) and minimum bactericidal concentration (MBC) tests. The strains were stored at −80 °C and were then cultured for 18 to 24 h on Tryptic Soy Agar (TSA) plates at 37 °C. A colony was seeded into Tryptic Soy Broth (TSB) and incubated over night at 37 °C. The inoculum was adjusted to 0.5 McFarland (1.5–2 × 10^8^ CFU mL^−1^) using a turbidimeter, and then diluted 1/10 to reach 1 × 10^7^ CFU mL^−1^. Preparation of 96-microwell plates was performed by the addition of 120 μL of Mueller-Hinton broth (MHB) in each well.

Preparation of 96-microwell plates was carried out by adding 120 μL of Mueller-Hinton broth (MHB) in each well. A zeolite stock was prepared by mixing 250 mg in 5 mL of sterile water. The first well received 120 μL of the zeolite stock along with 120 μL of MHB, resulting in an initial concentration of 7.81 mg mL^−1^. From this, 10 dilutions were prepared. Next, 5 μL of the 1 × 10^7^ CFU mL^−1^ inoculum was added, resulting in a final inoculum of 5 × 10^5^ CFU mL^−1^. Each microwell plate was incubated at 37 °C for 24 h. After incubation, 10 μL of MTT (3-(4,5-dimethylthiazol-2-yl)-2,5-diphenyltretazole bromide) was added to each microwell. It was then incubated for 30 min to observe the color change, and an MIC reading was taken. Wells that turned purple indicated bacterial growth (metabolically active bacteria). MTT was metabolically reduced because of the action of the electron transport chain. Non-viable bacteria or bacteria inhibited by an antibiotic will not induce a color change. Next, 2 μL of each sample was inoculated in TSA and incubated for 48 h. To determine the minimum bactericidal concentration (MBC) and avoid carry-over possibilities, MBC determination was performed by filtering on a 0.45-micron filter paper, washing with PBS, and subsequent transfer to MHA according to CLSI recommendations (M26-A 2011). Subsequently, the growth at 24 and 48 h was observed.

In a second step, the antibacterial activity of the modified zeolites against clinical methicillin-resistant strain *Staphylococcus aureus* [21] and carbapenem-resistant *Acinetobacter baummani* [22] strains were performed using agar diffusion assays. To perform the assays, a series of discs was developed with the modified zeolites at different concentrations. Discs were obtained using a brand and a mass due to 320 mg x-trip discs. This trial was repeated twice separately. Table 1 shows the different concentrations obtained between the different mixtures of the modified zeolites to learn the individual and synergistic effects of the antibacterial capacity of this biomaterial.

Then, agar plates were prepared with 25 mL of MHA. The bacterial inoculum of each strain was prepared in the same way as for the MIC test until 1 × 10^7^ CFU mL^−1^. A sterile torula was introduced into the inoculum until it was impregnated with the solution. Triplicates of each type of disc were exposed to UV light for 20 min per side and deposited on the agar, placed equidistant from each other and not less than 24 mm from center to center of the discs. With the plate closed, the tablets were allowed to adhere to the agar for about 15 min and then incubated for 16–18 h at 37 °C. The plates were read after 16–18 h of incubation. The inhibition halos of the tablets were read with the back of the meter foot under reflected light.

### 2.7. Statistical Analysis

For antimicrobial assays, an ANOVA test with multiple comparisons (Tukey’s method) was used, with 95% confidence intervals for calculating the differences. Differences were considered statistically significant at the 5% level. Data were analyzed using GraphPad Prism 6.0 (GraphPad Inc., La Jolla, CA, USA).

## 3. Results and Discussion

### 3.1. Characterization of ZLn and ZLnL

The chemical composition of ZLn was determined using quantitative wave scattering X-ray fluorescence spectrometry (XRF) [26]. The analysis revealed the following compositional ranges (see Table 2): SiO_2_ (63–66%) and Al_2_O_3_ (10–13%) as main components, along with significant amounts of Fe_2_O_3_ (1–4%) and of CaO (2–3%).There were smaller amounts Na_2_O, K_2_O, and MgO, and only traces of MnO, TiO_2_, and P_2_O_5_ were found. The percentage of moisture varied between 6 and 12%. The structural characterization was carried out using X-ray diffraction. The XRD diffraction pattern of the natural zeolite (see Appendix A) matched primarily with the crystalline phases of clinoptilolite and mordenite. However, some additional peaks corresponding to heulandite and quartz were also observed. The intensity of the peaks confirmed that the material was highly crystalline, with little to no amorphous phase present [27,28].

After completing the washing process with the hydrochloric acid solution 2.4 mol L^−1^, both the natural and the washed zeolite were characterized using various techniques (FTIR-ATR and BET, SEM). In Figure 1, the main characteristic bands attributed to zeolites are shown. These include the presence of hydroxyl groups in the region around 3440 cm^−1^, the presence of water molecules in the zeolite channels within the range of 2990 and 1600 cm^−1^, and the presence of asymmetric vibrational modes of Si–O and Al–O bonds at 1005 cm^−1^. These bands are indicative of the composition of the material, as well as the stretching vibrational modes of bonds between oxygen and elements such as silicon and aluminum.

Nitrogen gas adsorption tests were conducted at a temperature of 77 K to determine the specific surface area (Sg), pore volume (Vp), micropore volume (Vo), and mesopore volume (Vm). Prior to each measurement, the zeolite samples underwent degassing at a relatively low temperature (120 °C) for 3 h to remove water and any adsorbed gases that could affect the analysis. Table 3 presents the N_2_ adsorption–desorption isotherm values for the natural zeolite (ZLn) and the washed zeolite (ZLnL). The shape of the isotherms and the specific surface area (SBET) values obtained using the Brunauer–Emmett–Teller (BET) method, as well as the pore volume (VP) calculated from the amount of nitrogen adsorbed at a relative pressure of 0.95 atm, indicate that both samples are non-porous, without the presence of micropores (see Appendix A). However, it is evident that washing the zeolite helped improve the specific surface area and increased the pore volume. This improvement can be attributed to the removal of organic matter and impurities during the natural zeolite extraction process, as well as the elimination of a significant amount of iron salts. These results are consistent with those reported by Baghbanian et al., 2014, who observed an increase in surface area from 49.7 to 55.6 m^2^ g^−1^ after acid washing of natural clinoptilolite zeolite [4]. 

In addition, scanning electron microscopy (SEM) and energy dispersive X-ray spectroscopy (EDS) were used to morphologically characterize the natural and washed zeolites (ZLn and ZLnL). Figure 2A,B display the SEM images and EDS spectra of the ZLn and ZLnL, respectively. The SEM images of the ZLn and ZLnL show no significant structural differences when washed with the acid solution. However, the EDS analysis reveals the presence of the main elements associated with the zeolite-type structures, which aligns with previous results on clinoptilolite-type zeolite and its silicon–aluminum proportions [1,29]. Additionally, a decrease in iron content in the biomaterial is observed. Furthermore, based on the EDS values obtained, a slight decrease in the aluminum content can be observed compared to the natural zeolite. This decrease in the aluminum content is attributed to a decationization and dealuminization process [30].

### 3.2. Zeolite Saturation with Copper Ions

In Figure 3A,C, different adsorption studies of copper ions (II) in zeolite were conducted to determine the maximum ion retention capacity thought of cation exchange. The goal was to obtain a biomaterial with antimicrobial properties, such as using natural Chilean zeolites saturated with copper ions and subsequently compatible with CuO NPs. Figure 3A shows the study of the effect of the variation in zeolite. These studies are correlated with increasing doses, as observed in the data on exchange percentages. This correlation may be due to the increase in the surface area of the zeolite. However, for the purpose of this study, the saturation capacity was plotted to establishing the balance between the copper ions and the active sites in the zeolite. This plot reveals a different trend as the content of zeolite in the material increases, which can be explained by the overlapping adsorption sites [31]. 

Regarding saturation with varying initial concentrations of copper, this is an important study, as it helps determine the effective saturation capacity of the zeolitic material. In this case, a constant dose of ZLnL (2.0 g) was used while the concentration of copper ions was varied (see Figure 3B). It was demonstrated that the saturation capacity of the zeolite increased with an increase in the concentration of copper ions. This phenomenon is known as a concentration gradient, where increasing the concentrations exerts a force that overcomes the mass transfer of the copper ions in solution and the active sites present in the ZLnL. Similar phenomena have been observed in studies on the removal of contaminants in carbonaceous materials [32].

In the study of the variation of time (see Figure 3C), it is important to determine the optimal time for achieving maximum saturation and to minimize energy costs in the saturation processes. The results showed that maximum saturation was obtained at a time of 0.5 h, followed by a slight desorption due to the saturation of active sites in the zeolitic material. This information is valuable for reducing energy costs and can be used in the future scaling-up of the product. From these findings, it can be concluded that the zeolitic material achieved a copper concentration of 27,000 mg L^−1^, using a dose of 2.00 g of zeolitic material dispersed in a solution of copper ions at a concentration of 50.0 g L^−1^. The mixture was agitated at 200 rpm for a period of 0.5 h and at room temperature.

Appendix A displays the FTIR-ATR spectra of ZLnL compared with ZLnL in contact with different concentrations of Cu (ZLnLCu) at 10.0, 20.0, 30.0, and 50.0 g L^−1^, which represent the saturation processes. Initially, the presence of symmetric and asymmetric stretching signals of the zeolitic material can be verified in 3325, 1620, 1052, 793, 556, and 455 cm^−1^, which are characteristic of hydroxyl functional groups and vibrations of elements present in clinoptilolite-type zeolitic materials [4,33]. When the ZLnL is saturated with copper ions, a slight shift in the O–H stretching band is observed due to the Cu ion exchange process. Additionally, there are slight attenuated bands around 1452 and 898 cm^−1^, which can be attributed to exchange processes between sodium and copper ions [34,35]. These results are supported by SEM analysis, which shows the formation of copper salt layers on the ZLnL particles during the exchange process. This is further confirmed by EDS analysis, which indicates the presence of copper content in the ZLnL (see Figure 3D).

### 3.3. Copper Oxide Nanoparticles Supported on the Zeolite Surface

Figure 4 shows the TEM images and the SAED pattern for the ZLnLCuONPs samples, confirming that the copper oxide nanoparticles were supported on the surface of the ionically saturated zeolite to provide a higher antimicrobial concentration to the zeolite biomaterial. The SAED pattern showed the continuous rings that were separated from each other. The labeled rings, starting from the inner ring, were (1 1 0), (0 0 2), (1 1 1), (2 0 2), (0 2 0), and (0 2 2). The TEM and SAED results confirmed the successful support of the CuO nanoparticles [36,37]. 

The FTIR spectra of the ZLnL and the ZLnLCuONPs are shown in Appendix A. The FTIR spectra for the two samples have strong absorption bands in the 3700–3300 cm^−1^ range, which is due to the hydroxyl groups coordinating with the zeolite structure and corresponding to the water molecules. The vibrations within the lattice between 1300 and 450 cm^−1^ show slight shifts in the main absorption peak of the zeolite, possibly due to the presence of metal complexes or metal oxide nanoparticles. It can be observed that the spectrum of ZLnL (1080 cm^−1^) compared to that of the ZLnLCuONPs (1070 cm^−1^) appears shifted, which may be due to the interaction of the zeolite network and the copper oxide nanoparticles [38]. Although the vibrational peaks of the CuO NPs are not highlighted, the presence of nanoparticles in the lattice and their interaction with the zeolite lattice has led to changes in all the particular peaks of the zeolite, especially the main range (1080 cm^−1^), at lower wavelengths. On the other hand, the maximum intensity has been reduced in these areas [39].

### 3.4. Antibacterial Properties of the Biomaterial

The MIC of the tested zeolites against the clinical strain of *S. aureus* and the ATCC strain of *E. faecalis* were the same for ZLnL, with differences of at least five dilutions with respect to ZLnLCu and ZLnLCuONPs. The results are presented in Table 4 and Figure 5. In both strains, ZLnLCu and ZLnLCuONPs showed the same value for MBC, while for *S. aureus*, the MIC was lower compared to *E. faecalis*.

The previous research has described the process of obtaining natural Chilean zeolite saturated with copper ions and its antibacterial activity. Microbiological tests have demonstrated that zeolite saturated with copper salt ions (ZLnLCu) exhibits antimicrobial activity with bactericidal effect against strains of *Staphylococcus aureus* and ATCC *Enterococcus faecalis* 29212, with a MIC of 0.24 and 0.25 mg mL^−1^, respectively. This study has the advantage of not only using a control strain but also demonstrating the antimicrobial activity against common multi-resistant clinical strains in the dental biomedical field [25,40]. The results show the bactericidal effect of the biomaterial and its potential application as a filler in dental materials. 

Vergara-Figueroa et al. [41] reported the antibacterial activity of nanometer-sized zeolites doped with copper ions against methicillin-resistant *S. aureus*. The results suggested that the MIC of the antimicrobial agent used to inhibit bacterial growth was 0.75 mg mL^−1^, indicating that our biomaterial inhibits bacterial growth with a concentrations two times lower than the one reported. On the other hand, secondary endodontic infections frequently show a high prevalence of *Enterococcus faecalis*, as they can persist after root canal treatment [42]. Therefore, this biomaterial holds the promise of being highly useful in this field. Furthermore, there is limited information available on zeolites with proven antibacterial properties against this particular bacterial strain. Malek et al. [43] studied the antibacterial activity of copper-exchanged zeolite and zeolite synthesized from rice husk ash using the disk diffusion technique. The results indicated that zeolite exchanged with 900 ppm copper exhibited a bactericidal effect against *E. faecalis* with an inhibition halo of 2.25 mm. Although the results were obtained using different techniques, they provide evidence for the effectiveness of the new ZLnLCu material can be evidenced.

Zone of inhibition values were obtained for the different types of zeolites tested against the Gram-positive clinical strain of *S. aureus* (Figure 6A) and the Gram-negative clinical strain of *A. baumannii* (Figure 6B). The results are presented as mean values in Table 5. Both ZLnLCu and ZLnLCuONPs showed antibacterial activity against Gram-negative and Gram-positive bacteria. However, ZLnLCuONPs showed smaller inhibition halos compared to ZLnLCu, indicating that the addition of CuO nanoparticles does not affect the bactericidal activity of the material but does have an impact on other factors, such as production cost.

In this study, it was shown that ZLn does not have antimicrobial activity. In addition, ZLnLCu at the same concentration exhibited better activity than the ZLnLCuONPs for both the Gram-positive and Gram-negative strains, with Gram-positive bacteria being more sensitive. This finding is consistent with the results reported by Vergara-Figueroa et al. [41], who classified the bacteria as susceptible based on an average halo size of over 23 mm according to the CLSI [44]. The results presented in this study were obtained from two separate tests, each conducted in triplicate. According to the aforementioned cut-off point, both *S. aureus* and *A. baumannii* were found to be susceptible to saturated zeolite from 27,000 mg mL^−1^, which is equivalent to 27 g mL^−1^ [41]. Additionally, the superior antibacterial efficacy of ZLnLCu has also been attributed to the large surface area and small particle size of ZLnL [19].

## 4. Conclusions

Based on the result, the saturation capacity of the zeolite was determined by adding copper ions and incorporating copper oxide nanoparticles to assess its antibacterial activity. By evaluating different experimental conditions, a copper ion saturation of 27.0 g L^−1^ was achieved through an ion exchange process, which was confirmed by atomic adsorption and evidenced by EDS analysis. Furthermore, the TEM technique revealed the presence of copper oxide nanoparticles supported on the surface of the zeolite. Through microbiological tests, it was demonstrated that copper-ion-modified zeolites/nanostructures can serve as effective antibacterial agent for dental biomedical applications. However, it is crucial to conduct further research on the toxicological aspects of these materials to validate these findings.

## Figures and Tables

**Figure 1 nanomaterials-13-02140-f001:**
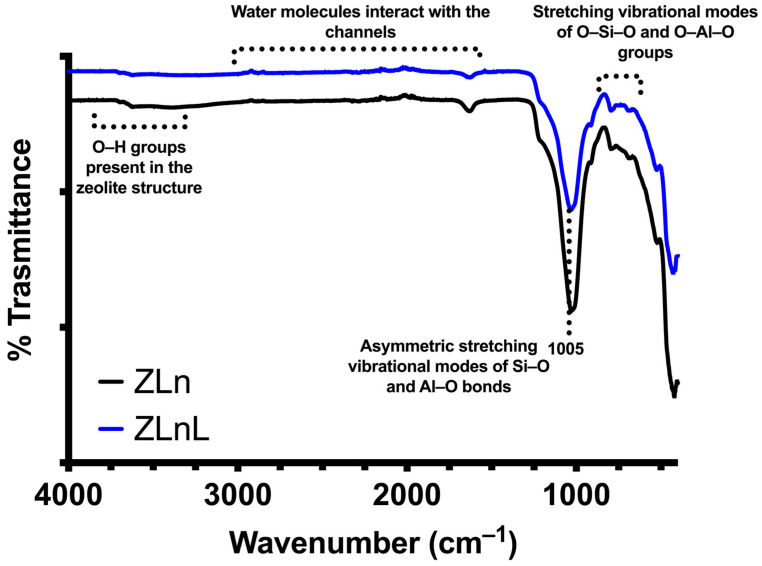
Characterization of natural zeolite (ZLn) by FTIR before and after washing with hydrochloric acid (ZLnL).

**Figure 2 nanomaterials-13-02140-f002:**
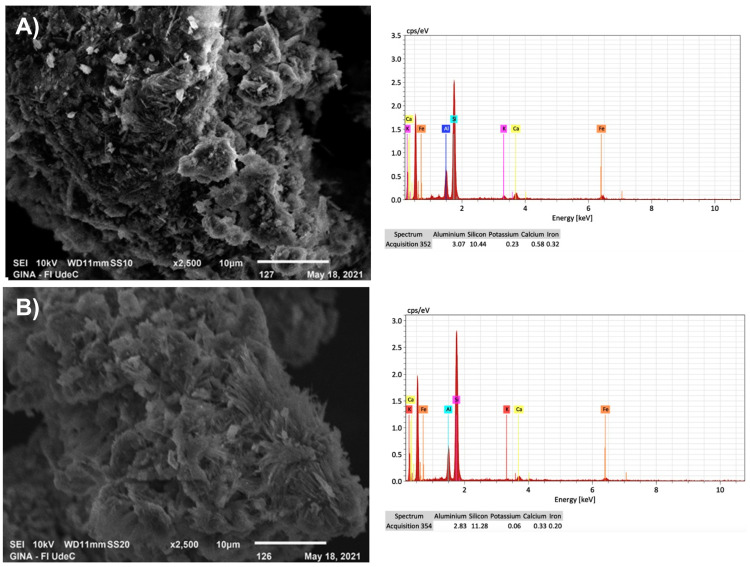
SEM image and EDS spectra of (**A**) ZLn and (**B**) ZLnL.

**Figure 3 nanomaterials-13-02140-f003:**
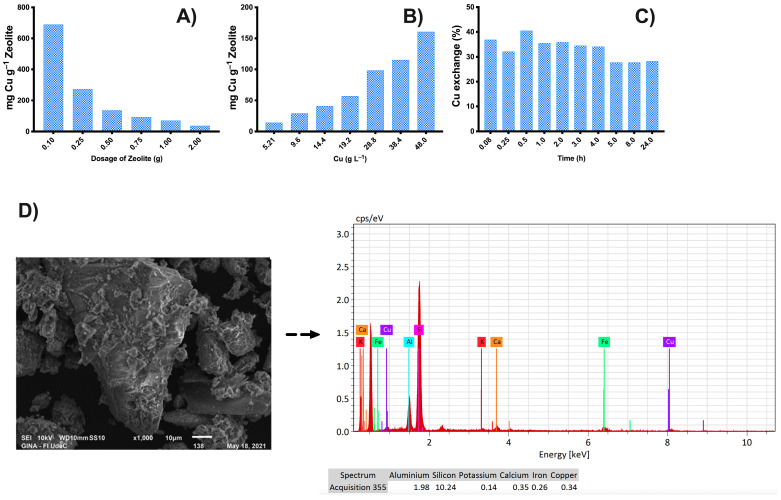
Study of the saturation capacity of zeolite with copper ions. (**A**) Variation in the zeolite dose, (**B**) variation in the concentrations of copper ions, (**C**) study of the saturation capacity over time, and (**D**) SEM image and EDS spectrum of zeolite saturated with copper ions.

**Figure 4 nanomaterials-13-02140-f004:**
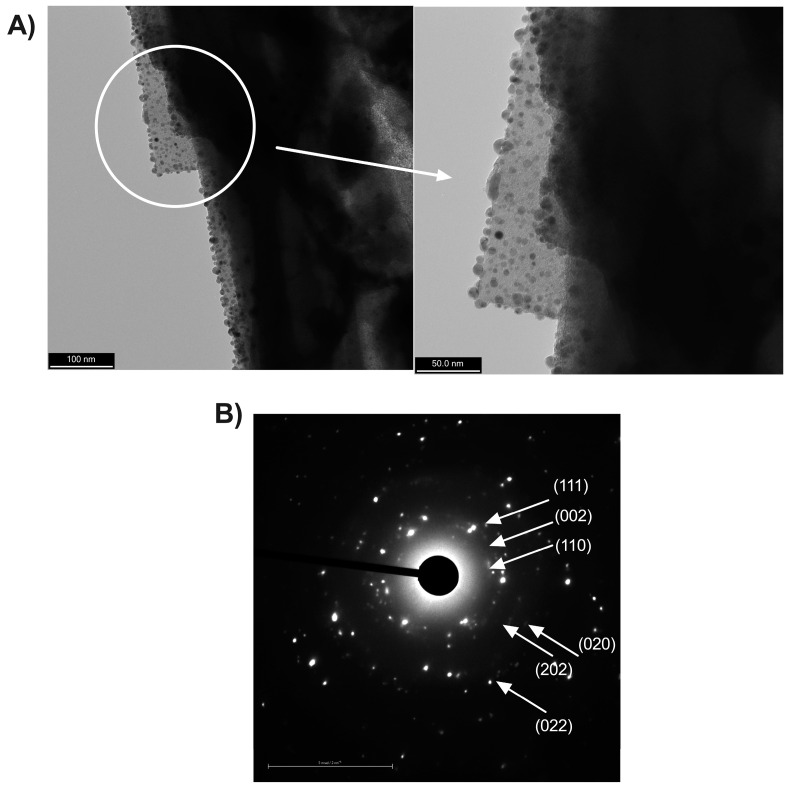
(**A**) TEM image and (**B**) corresponding SAED pattern of the ZLnCuONPs.

**Figure 5 nanomaterials-13-02140-f005:**
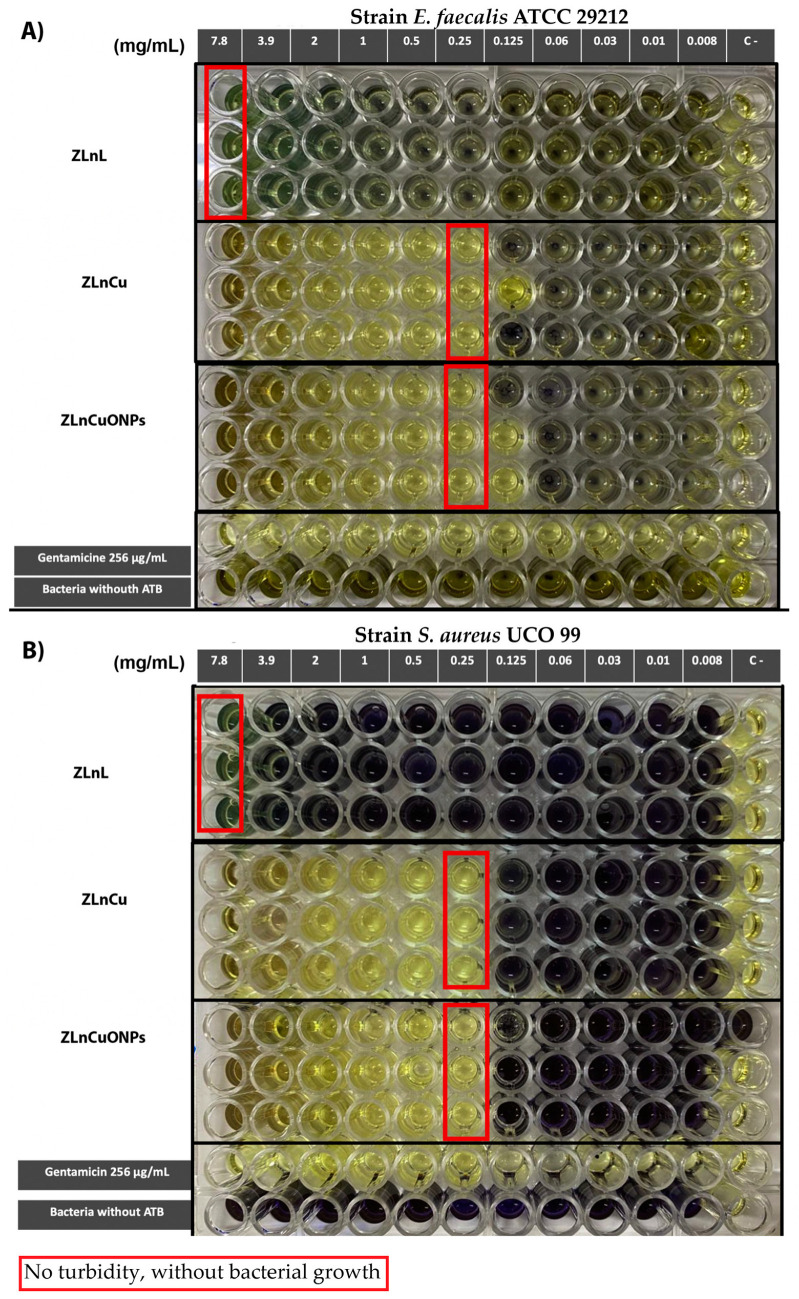
Broth dilution test: (**A**) MIC of natural zeolites and modified zeolites against *E. faecalis*. (**B**) MIC of natural zeolites and modified zeolites against *A. baumannii*.

**Figure 6 nanomaterials-13-02140-f006:**
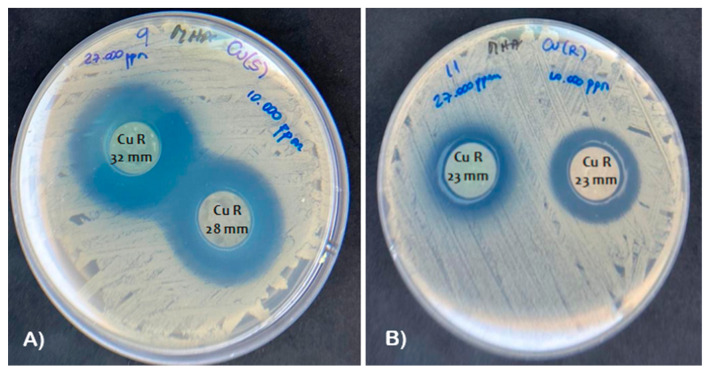
Zone of inhibition of ZLnCu tested against (**A**) methicillin-resistant Gram-positive clinical strain of *S. aureus* and (**B**) carbapenem-resistant Gram-negative clinical strain of *A. baumannii*.

**Table 1 nanomaterials-13-02140-t001:** Zeolite disc concentrations with the presence of copper ions and NPS for antimicrobial studies.

Tablets	Copper Concentration (mg L^−1^)		
	ZLnLCu	ZLnLCu + NPs	ZLnLNPs
1	-	100,000	63,000
2	-	50,000	50,000
3	27,000	27,000	27,000
4	10,000	10,000	10,000
5	5000	5000	5000
6	1000	1000	1000

Note: In all the tests, a white tablet was made that corresponded to washed zeolite without NPs and copper ions.

**Table 2 nanomaterials-13-02140-t002:** Chemical composition and physical properties of the zeolite used.

Compounds	Composition%
SiO_2_	64.19
TiO_2_	0.51
Al_2_O_3_	11.65
Fe_2_O_3_	2.53
MnO	0.03
MgO	0.66
CaO	3.42
Na_2_O	0.75
K_2_O	1.60
**Physical Properties**	
Cation exchange capacity (CIC)	86.82 a 112.88 cmol/Kg
Thermal stability	<450 °C
Chemical stability (pH)	8.9
Main zeolitic component	Clinoptilolite–Mordenite.
Chemical name	Calcium potassium magnesium aluminosilicates hydrated

**Table 3 nanomaterials-13-02140-t003:** Determination of specific surface and pore volume.

Sample	SBET m^2^ g^−1^	Pore Volume cm^3^ g^−1^
ZLn	36.2	0.044
ZLnL	58.6	0.054

**Table 4 nanomaterials-13-02140-t004:** Mean values of minimum inhibitory concentration (MIC) and minimum bactericidal concentration (MBC) of modified zeolites through serial dilution test.

Bacterial Strain	MIC (mg mL^−1^) ± SD			MBC (mg mL^−1^) ± SD		
	ZLnL	ZLnLCu	ZLnLCuONPs	ZLnL	ZLnLCu	ZLnLCuONPs
*E. faecalis* ATCC 29212	7.8 ± 0.5	0.25 ± 0.05 *	0.25 ± 0.01 *	7.8 ± 0.5	1.12 ± 0.9 *	3.9 ± 0.05
*S. aureus* UCO 99	7.8 ± 0.9	0.24 ± 0.01 *	0.24 ± 0.02 *	7.8 ± 0.9	1.12 ± 0.9 *	3.9 ± 0.05

The values are represented as the average of triplicate MIC and MBC values and their respective standard deviation. * Mean values with significant differences compared with ZLnL. (*p* < 0.05).

**Table 5 nanomaterials-13-02140-t005:** Average inhibition zone (mm) and standard deviation for modified zeolites at different concentrations.

Bacterial Strain	ZLnCu Inhibition Zone (mm)				ZLnCuONPs Inhibition Zone (mm)						Control	
	Concentrations (g ** L^−1^)**										ZLnL (−)	ATB (+)
	1	5	10	27	1	5	10	27	50	100		
*S. aureus* UCO 99	21.0 ± 3.5 ^B^	27.1 ± 2.6 ^A^	32.5 ± 1.6 * ^A^	38.5 ± 1.1 * ^A^	20.5 ± 3.3 ^B^	21.1 ± 3.6 ^B^	24. ± 3.2 ^A^	28.8 ± 2.9 * ^A^	32.6 ± 1.5 * ^A^	36.8 ± 0.7 * ^A^	19.5 ± 2.9	21.0 ± 0.3
*A. baumannii* UCO 538	13.0 ± 0.0 ^C^	13.0 ± 0.0 ^C^	16.5 ± 0.5 ^C^	26.6 ± 0.4 * ^A^	13.0 ± 0.0 ^C^	13.0 ± 0.0 ^C^	13.0 ± 0.0 ^C^	14.0 ± 0.0 ^C^	19.6 ± 0.2 ^C^	25.0 ± 0.3 * ^A^	13.0 ± 0.0	21.0 ± 0.3

Abbreviations: ZLnLCu, cooper ion-saturated zeolite; ZLnLCuONPs, zeolite supported with copper oxide nanoparticles; ZLnL, washed natural zeolite. (^A^) Susceptible to antibacterial material. (^B^) Intermediate. (^C^) Not susceptible to antibacterial material. * Mean values with significant differences (*p* < 0.05).

## Data Availability

The raw/processed data required to reproduce these findings cannot be shared at this time as the data are also part of an ongoing study.

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
