# Peer review of "Study of the Antibacterial Capacity of a Biomaterial of Zeolites Saturated with Copper Ions (Cu2+) and Supported with Copper Oxide (CuO) Nanoparticles"

_nanomaterials, 2023, doi:10.3390/nano13142140_

Round 1
Reviewer 1 Report
The main question addressed by the research of Romero et al relates to the demonstration that a zeolite-based biomaterial incorporating copper ions has a bactericidal effect against Gram-positive and -negative bacteria.
The topic is not particularly original as much work on the biocidal effects of copper has been published, however, the authors zeolite is the most abundant type and therefore is easily sourced. Nevertheless, the authors need to explain clearly the advantages of their copper-doped zeolite over others in the literature.
The methodology is sound throughout. The analytical chemistry employs appropriate methods allowing the authors to draw valuable conclusions concerning the composition of the materials and their morphology. The microbiological work uses standard MIC and disk diffusion methods to demonstrate greater efficacy than had been reported recently by Vergara-Figueroa for a very similar system derived from the same Chilean zeolite. I would like to see a rationale for testing against E. faecalis particularly considering the Vergara-Figueroa group tested on standard Gram-negative E. coli and Gram-positive S. aureus. The authors note that “control strains that are usually sensitive to various antimicrobials” so it would have been interesting to have the materials tested on more resistant organisms. If the strains used are expected to be susceptible to antimicrobials, it is not surprising that positive results were obtained.
I recommend that the authors demonstrate distinctiveness from the work of Vergara-Figueroa by testing their zeolites additionally against E. coli.
The conclusions are consistent with the evidence presented and successfully demonstrate that the copper doped zeolite functions as an effective antimicrobial material while the undoped zeolite and, perhaps surprisingly, the doped material incorporating CuO nanoparticles did not.
The references are appropriate and relate to very relevant literature sources. The tables and figures are clear and represent the data well.
There are numerous examples of poor English which need to be addressed, for example:
“In the search for applications at the microbiological level, it has been necessary to they have realized both materials at the organic and inorganic level as are clayey materials”
This is a confusing, incomplete sentence. Who are ‘they’?
“ Yao et al., Study the antimicrobial capacity of Zeolites “ includes unnecessary capitalisation.

Some minor checks of the English is needed. I suggest it is read by a native English speaker who will quickly identify the errors.
Reviewer 2 Report
The objective of this investigation is focused on obtaining an antimicrobial biomaterial from a saturated natural zeolite with Cu2+ ions and supported with copper oxide nanoparticles, taking advantage of the high surface advantage of the zeolite presents that will allow the dispersion of CuO nanoparticles and antibacterial performance. This is a very interesting work and can be considered for publication. The following comments should be addressed;
1. Formatting and expressions errors in the manuscript should be corrected, such as, two expressions of g L-1 and g/L, etc.
2. Is it possible to determine whether the copper oxide is copper oxide or cuprous oxide? This is important for reaction. An XRD test is recommended.
3. Copper ion or copper oxide, which plays an antibacterial activity?
4. It is recommended to provide higher quality pictures, some picture fonts are not clear.
The objective of this investigation is focused on obtaining an antimicrobial biomaterial from a saturated natural zeolite with Cu2+ ions and supported with copper oxide nanoparticles, taking advantage of the high surface advantage of the zeolite presents that will allow the dispersion of CuO nanoparticles and antibacterial performance. This is a very interesting work and can be considered for publication. The following comments should be addressed;
1. Formatting and expressions errors in the manuscript should be corrected, such as, two expressions of g L-1 and g/L, etc.
2. Is it possible to determine whether the copper oxide is copper oxide or cuprous oxide? This is important for reaction. An XRD test is recommended.
3. Copper ion or copper oxide, which plays an antibacterial activity?
4. It is recommended to provide higher quality pictures, some picture fonts are not clear.
Reviewer 3 Report
(1) The abstract of this manuscript should be rewritten. There is no need for such a comprehensive and detailed description of the experimental methods section, as these experimental methods are common and mature tools. The abstract should include the main results, viewpoints, and focus on describing the innovation of the research work.
(2) The EDS spectrum in Figure 3 is not clear, please use a high-resolution image.
(3) The resolution of Figure 3D (FTIR spectra) should be increased and set as a new Figure 4 separately. Currently, it is difficult to see the changes described by the author from this image.
(4) For Figure 4, the Electron Diffusion Pattern or EDS analysis results should be supplemented, otherwise how can we confirm that these nanoparticles are CuO or Ag2O?
(5) Line 339, (P<.05) should be (P<0.05).
(6) The author should supplement more intuitive antibacterial experiments, such as the antibacterial circles of different samples.
(7) The FTIR-ATR spectrum of ZLnCuONPs should be supplemented.
(8) The manuscript lacks detailed interpretation of many key issues. For example, what is the presence state of copper ions in ZLnCu samples? Has it entered the crystal structure of zeolite?
(9) What are the copper salts deposited on the surface of zeolite particles? Is the sediment in Figure 3e Copper(II) sulfate? If Copper(II) sulfate plays an antibacterial role, what role does zeolite play (can zeolite not be used)?
(10) The author claims to have conducted a BET experiment, why is there no corresponding experimental results (adsorption curve) in the manuscript?
(11) The state of art of this manuscript is relatively weak. Please supplement the latest research literature:
a. Lei S C, Qi Y M*, Zhao L C, An H L, Qu C, Wang X, Wang G J, Cui C X, Shen Y T. Synergistic effect of Ag and Cu on improving in vitro biological properties of K2Ti6O13 nanowires for potential biomedical applications[J]. Biomed Mater, 2023, 18(2):025024
b. Weiwei Zhang, Xin Wang, Yuanhui Ma, Haoran Wang, Yumin Qi*, Chunxiang Cui*. Enhanced photocatalytic and antibacterial activities of K2Ti6O13 nanowires induced by copper doping. Crystals 2020, 10, 400.
There are many issues with English writing in this article. The reviewer suggests that the author use language polishing services or ask native English speakers to make careful revisions. The following issues (not limited to) need to be modified:
(1) Line39-42, this sentence is too long, please break it into several sentences. Different applications should be revised to different fields.
(2) Line58-59,“it has been necessary to they have realized…”,Obviously there is a Syntax error here.
(3) Line66, “Rodriguez-Méndez et al., He studied..” Why arrange two subjects in a sentence?
(4) How are the physical properties in Table 2 obtained? References should be provided.
(5) Line 252, why start this paragraph with 'However'? This is illogical.
(6) Figure 2. SEM image and EDS spectrum, (a) ZLn and (b) ZLnL. should be “Figure 2. SEM image and EDS spectrum of (a) ZLn and (b) ZLnL.”
Round 2
Reviewer 3 Report
1)Is there only diffraction information of CuO in Figure 4B without zeolite? Is zeolite an amorphous structure or a single crystal structure? Can the XRD spectrum be provided?
2)The questions raised by the reviewer seem to have received some response, but the language description still needs further improvement.
The questions raised by the reviewer seem to have received some response, but their language description still needs further improvement.
Author Response
Consulte el archivo adjunto.
